# Tumor suppressive role of sestrin2 during colitis and colon carcinogenesis

Seung-Hyun Ro[1,2†], Xiang Xue[1†], Sadeesh K Ramakrishnan[1], Chun-Seok Cho[1], Sim Namkoong[1], Insook Jang[1], Ian A Semple[1], Allison Ho[1], Hwan-Woo Park[1,3,4], Yatrik M Shah[1*], Jun Hee Lee[1*]

[1]Department of Molecular and Integrative Physiology, University of Michigan, Ann Arbor, United States; [2]Department of Biochemistry, University of Nebraska, Lincoln, United States; [3]Department of Cell Biology, College of Medicine, Konyang University, Daejeon, Republic of Korea; [4]Myung-Gok Eye Research Institute, Konyang University, Seoul, Republic of Korea

**Abstract** The mTOR complex 1 (mTORC1) and endoplasmic reticulum (ER) stress pathways are critical regulators of intestinal inflammation and colon cancer growth. Sestrins are stress-inducible proteins, which suppress both mTORC1 and ER stress; however, the role of Sestrins in colon physiology and tumorigenesis has been elusive due to the lack of studies in human tissues or in appropriate animal models. In this study, we show that human *SESN2* expression is elevated in the colon of ulcerative colitis patients but is lost upon p53 inactivation during colon carcinogenesis. In mouse colon, Sestrin2 was critical for limiting ER stress and promoting the recovery of epithelial cells after inflammatory injury. During colitis-promoted tumorigenesis, Sestrin2 was shown to be an important mediator of p53's control over mTORC1 signaling and tumor cell growth. These results highlight Sestrin2 as a novel tumor suppressor, whose downregulation can accelerate both colitis and colon carcinogenesis.

**\*For correspondence:** shahy@ umich.edu (YMS); leeju@umich. edu (JHL)

[†]These authors contributed equally to this work

**Competing interests:** The author declares that no competing interests exist.

## Introduction

Colorectal carcinoma (CRC) is an important contributor to cancer mortality and morbidity. The lifetime risk of developing CRC in the US is 4–5%, and approximately one-third of CRC patients die from the disease (*Siegel et al, 2015*). Although the pathogenetic mechanisms underlying CRC development are complex and heterogeneous, several critical genes and pathways important in its initiation and progression are well characterized, such as Wnt-APC, Ras-MAPK, p53 and DNA repair pathways (*Fearon, 2011*). In addition to these components, mammalian target of rapamycin complex 1 (mTORC1), a protein kinase that is essential for cell growth (*Hay and Sonenberg, 2004*; *Zoncu et al, 2011*), was recently found to play a key tumorigenic role during CRC development induced by either colitis (*Thiem et al, 2013*) or a genetic mutation (*Faller et al, 2015*; *Hardiman et al, 2014*).

In addition to promoting cancer cell growth, mTORC1 hyperactivation can lead to unrestricted protein synthesis, resulting in the accumulation of unfolded protein, endoplasmic reticulum (ER) stress and tissue injury (*Ozcan et al, 2008*; *Park et al, 2014*; *Young et al, 2013*), which together can contribute to tumor progression (*Wang and Kaufman, 2014*). ER stress has been shown to be critically involved in the pathogenesis of colitis and colon inflammation (*Bertolotti et al, 2001*; *Cao et al, 2013*; *Kaser et al, 2008*), which is an important risk factor of CRC development (*Thorsteinsdottir et al, 2011*) and a well-characterized tumor promoter (*Grivennikov et al, 2012*; *Terzic et al, 2010*). Mechanisms of how the mTORC1 and ER stress signaling pathways are

**eLife digest** An organ that is inflamed has an increased risk of developing cancer. Inflammation can be elicited in various ways; and intestinal inflammation and colon cancer development are often associated with a protein complex – called mTORC1 – being overactive in the tissue. This protein complex has been studied in other contexts and is known to instruct cells to produce more proteins. However, when too much protein is made too quickly, cells cannot carry out their routine quality checks. This, in turn, can lead to unfolded proteins accumulating in the cell, which is stressful and damaging, and can cause inflammation. Increased production of proteins and other biomolecules can also allow the uncontrolled growth of cancer cells.

Other recently discovered proteins – called sestrins – can counteract the cancer-promoting effects of overactive mTORC1. Sestrins achieve this via several mechanisms, but as yet almost nobody had studied the role of these proteins in intestinal inflammation and colon cancer.

Ro, Xue et al. deleted the genes for two members of the sestrin family, called Sestrin2 and Sestrin3, in mice and showed that their colons were more prone to inflammation. Additional analysis showed that people with ulcerative colitis – a condition in which the colon is chronically inflamed – have elevated levels of Sestrin2, whereas very low levels of Sestrin2 could be detected in tissue samples from patients with colon cancers. These data suggested that Sestrin2 might be trying to protect cells from injury and acts as a barrier to cancer formation.

Ro, Xue et al. then used biochemical techniques in human cancer cells grown in the laboratory to show that Sestrin2 inhibits mTORC1, making these cells grow less. Colon cancer cells with little or no Sestrin2 were also more resistant to chemotherapy than control cells with normal levels of Sestrin2. Lastly, a type of colon cancer that is associated with inflammation grew faster in mice that lacked the gene for Sestrin2. Taken together these findings represent evidence that Sestrin2 acts as a tumor suppressor in the colon. Future experiments might investigate how losing Sestrin2 makes these cells more resistant to chemotherapy and whether sestrins act as tumor suppressors in other tissues as well.

regulated in the colon, especially during colon injury, inflammation and tumorigenesis, are poorly understood.

Sestrins are a family of stress-inducible proteins that are widely conserved throughout animal species (*Lee et al, 2013*). Sestrins were originally identified as a target of the tumor suppressor p53 (*Budanov et al, 2002*; *Velasco-Miguel et al, 1999*). Sestrins have two important functions, suppressing reactive oxygen species (ROS) (*Budanov et al, 2004*) and inhibiting mTORC1 (*Budanov and Karin, 2008*). The ROS-suppressing effect of Sestrins is dependent, at least partially, on mTORC1 inhibition, which promotes autophagic degradation of dysfunctional mitochondria or an Nrf2 inhibitor Keap1 (*Bae et al, 2013*; *Lee et al, 2010*; *Woo et al, 2009*). However, Sestrin can also function as an active oxidoreductase that can directly detoxify ROS such as alkylhydroperoxides (*Kim et al, 2015a*). Sestrins inhibit mTORC1 through the activation of AMP-activated protein kinase (AMPK) and the subsequent inactivation of Rheb GTPases (*Budanov and Karin, 2008*; *Sanli et al, 2012*). Independently of AMPK, Sestrins can also inhibit Rag GTPases (*Chantranupong et al, 2014*; *Kim et al, 2015b*; *Parmigiani et al, 2014*; *Peng et al, 2014*), which are essential for mTORC1 activity. Sestrin-mediated inhibition of mTORC1 is also critical for limiting protein synthesis upon unfolded protein accumulation (*Bruning et al, 2013*; *Park et al, 2014*) or amino acid starvation (*Peng et al, 2014*; *Wolfson et al, 2015*; *Ye et al, 2015*), thereby suppressing ER stress or nutrient crisis.

In light of these important cellular functions, the present study assessed if Sestrin functions as a coordinator of mTORC1 and ER stress signaling pathways in the colon during intestinal inflammation and carcinogenesis. Our data collected from patient samples, mouse models of colitis and colitis-associated cancer, cultured colon cancer cell lines as well as data mining of large-scale transcriptome analyses, concertedly indicate that Sestrin2, a member of the Sestrin family, is important for proper regulation of mTORC1 and ER stress pathways during colon injury, and thereby functions as a suppressor of colitis and colon cancer development.

## Results

### Loss of Sestrin2 sensitizes mice to colon injury

Increased ER stress and excessive ROS accumulation are hallmarks of colon inflammation and colitis (*Fritz et al, 2011*; *Zhu and Li, 2012*). Sestrins can be induced upon either of these stresses and are critical to dampen their detrimental consequences (*Bruning et al, 2013*; *Lee et al, 2013*; *Park et al, 2014*). Therefore, to understand the role of Sestrins in colitis, expression of Sestrins was analyzed in tissues isolated from patients with ulcerative colitis (UC). Sestrin1 mRNA (*SESN1*) was unaltered in UC (*Figure 1A*); however, expression of *SESN2* (*Figure 1B*) and *SESN3* (*Figure 1C*) was significantly increased in the intestine of patients with UC.

To examine whether colitis-induced Sestrin2 and Sestrin3 play a physiological role in maintaining intestinal homeostasis, WT and *Sesn2^{-/-}/Sesn3^{-/-}* mice were treated with dextran sulfate sodium (DSS) in the drinking water to induce colitis. DSS treatment for 7 days led to substantial weight loss in both WT and *Sesn2^{-/-}/Sesn3^{-/-}* mice (*Figure 1—figure supplement 1A*). After placing back on regular water, WT mice recovered their body weight (*Figure 1—figure supplement 1A*). However, *Sesn2^{-/-}/Sesn3^{-/-}* mice did not show any recovery and continued to lose body weight until the experimental endpoint (5 days during the recovery phase; *Figure 1—figure supplement 1A*). *Sesn2^{-/-}/Sesn3^{-/-}* mice also showed a dramatic decrease in colon length when compared to WT mice (*Figure 1—figure supplement 1B*), indicative of strongly exacerbated DSS-induced colitis. Histological examination of colon tissue sections also revealed significant epithelial degeneration in *Sesn2^{-/-}/Sesn3^{-/-}* mice following the 5 days of recovery from the 7-day DSS treatment, while WT mice exhibited substantial regeneration of epithelial structure at the same time point (*Figure 1—figure supplement 1C*). The increased susceptibility of *Sesn2^{-/-}/Sesn3^{-/-}* mice to DSS-induced injury (*Figure 1—figure supplement 1A–C*) was recapitulated in *Sesn2^{-/-}* mice; although both WT and *Sesn2^{-/-}* mice develop severe colitis with one week of DSS treatment (*Figure 1D* and *Figure 1—figure supplement 2*), WT mice successfully recovered from injury after one additional week of regular water treatment, while *Sesn2^{-/-}* mice did not (*Figure 1D–F*). These results demonstrate a critical role for Sestrin2 in restoring intestinal homeostasis after injury.

### Sestrin2-deficient mice fail to recover from DSS-induced colitis

We examined molecular markers for cell death and inflammation in the colons of WT, *Sesn2^{-/-}* and *Sesn2^{-/-}/Sesn3^{-/-}* mice after DSS treatment. At 5 days after DSS injury, WT mice displayed a very small number of apoptotic cells (*Figure 1G* and *Figure 1—figure supplement 1D*), consistent with the histological observation showing that the colon epithelium had been restored (*Figure 1F* and *Figure 1—figure supplement 1C*). However, a significant number of apoptotic cells were observed in the colon epithelium of both *Sesn2^{-/-}* and *Sesn2^{-/-}/Sesn3^{-/-}* mice (*Figure 1G* and *Figure 1—figure supplement 1D*), consistent with the degenerative phenotypes observed in these mice. Proliferating cell nuclear antigen (PCNA) staining of WT colon displayed a normal pattern of cell proliferation; PCNA staining is confined to the base of colon crypts in WT mice (*Figure 1H* and *Figure 1—figure supplement 1E*), where epithelial progenitor cells are undergoing homeostatic proliferation that maintains normal turnover of the epithelium. However, the colon epithelium of both *Sesn2^{-/-}* and *Sesn2^{-/-}/Sesn3^{-/-}* mice exhibited an increased number of PCNA-positive cells throughout the degenerated epithelium (*Figure 1H* and *Figure 1—figure supplement 1E*). This result suggests that, in order to compensate for the apoptotic loss of epithelial cells, colonocytes of both *Sesn2^{-/-}* and *Sesn2^{-/-}/Sesn3^{-/-}* mice were undergoing active proliferation. Immunohistochemical staining of macrophage marker F4/80 (*Figure 1I* and *Figure 1—figure supplement 1F*), as well as quantitative RT-PCR examination of inflammation markers *Tnfa* (*Figure 1J*), *Il6* (*Figure 1K*), *Il1b* (*Figure 1L*) and *Il10* (*Figure 1M*), show that *Sesn2^{-/-}* mice had increased the levels of colon inflammation after DSS injury. These data collectively indicate that Sestrin2 deficiency exacerbates DSS-induced colon damage and inflammation.

### Sestrin2 expression in the extra-hematopoietic compartment suppresses colitis

Inflammatory cytokine signaling instigated by bone marrow-derived immune cells, such as macrophages, is known to be important for the progression of colitis as well as colon cancer (*Terzic et al,*

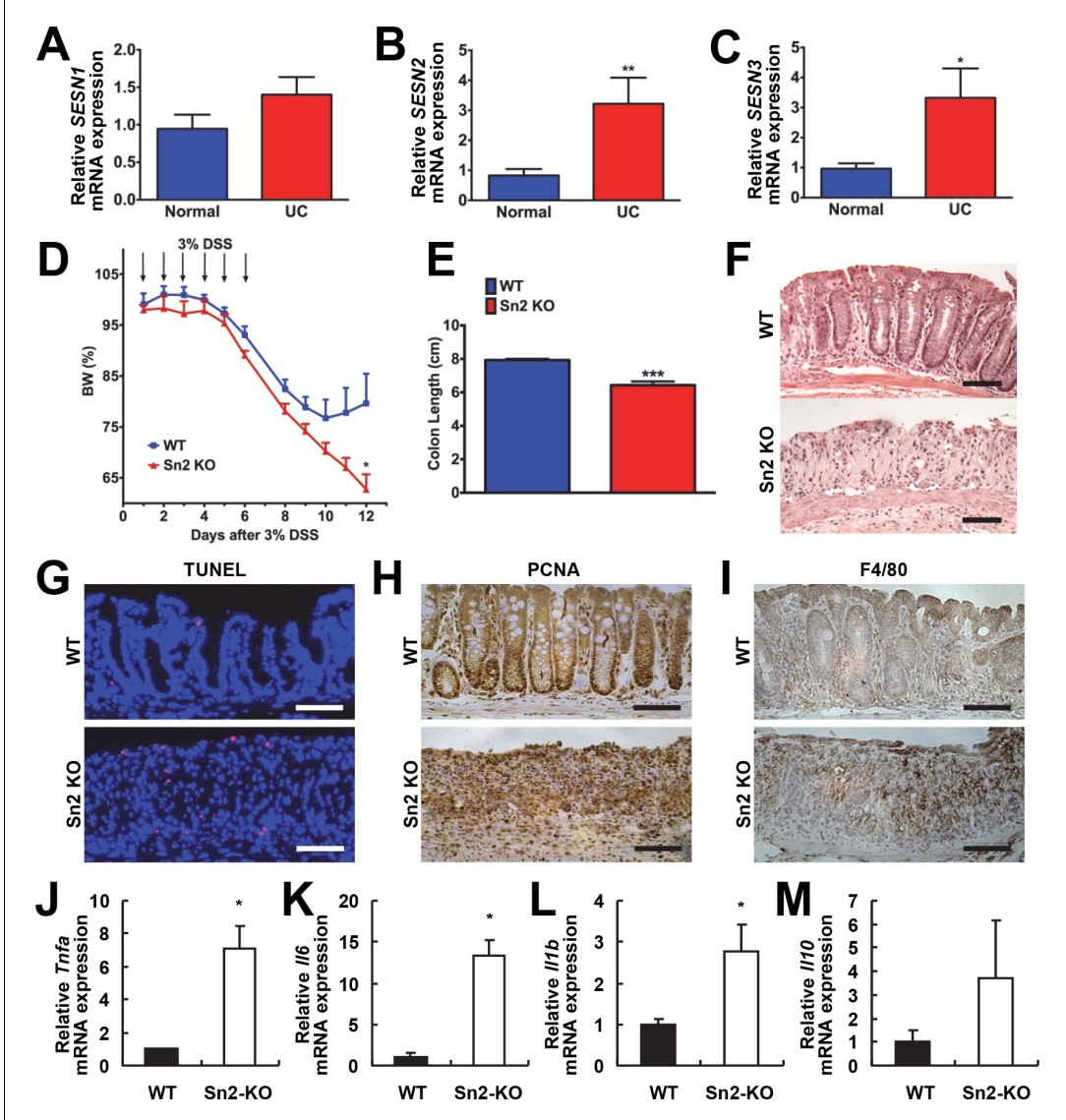

**Figure 1.** Protective function of Sestrin2 against colon injury. (**A-C**) Upregulation of human *SESN2* and *SESN3* expression in ulcerative colitis (UC). mRNA expression of human *SESN1-3* was analyzed through quantitative RT-PCR of non-inflamed (Normal) and inflamed (UC) colon tissues from patients with UC (n=10; mean ± s.e.m.). These samples were histologically confirmed and formerly described (*Xue et al, 2013*). (**D-M**) Loss of Sestrin2 impairs recovery from DSS-induced colitis in mice. 6-month-old WT and *Sesn2⁻/⁻* mice (n=4 each) were treated with 3% DSS in drinking water for 6 days (arrows), followed by 6 days of regular water. Body weight was measured over 12 days (**D**; mean ± s.e.m.). At the final day of the experiment, mice were sacrificed and colon length was measured (**E**). The data are shown as the mean ± s.e.m. The colons were isolated and fixed for H&E staining (**F**), TUNEL staining (**G**), PCNA staining (**H**) and F4/80 staining (**I**). The levels of the indicated mRNAs, which are indicative of active inflammation, were quantified by real-time PCR (**J-M**; mean ± s.e.m.). *p<0.05, **p<0.01, ***p<0.001. *P* values are from Student's t-test. Scale bars, 100 μm.

The following figure supplements are available for figure 1:

**Figure supplement 1.** Hypersensitivity of *Sesn2⁻/⁻/Sesn3⁻/⁻* mice against DSS-induced colon injury.

**Figure supplement 2.** Acute colon injury is comparable between WT and *Sesn2⁻/⁻* mice during DSS treatment.

*2010*). We examined whether the expression of Sestrin2 in the bone marrow-derived hematopoietic compartment is important for the protective role of Sestrin2 in colitis. For this purpose, reciprocal bone marrow chimera experiments were performed: WT bone marrow was introduced into lethally irradiated *Sesn2⁻/⁻* mice (WT→Sn2) while *Sesn2⁻/⁻* bone marrow was introduced into lethally

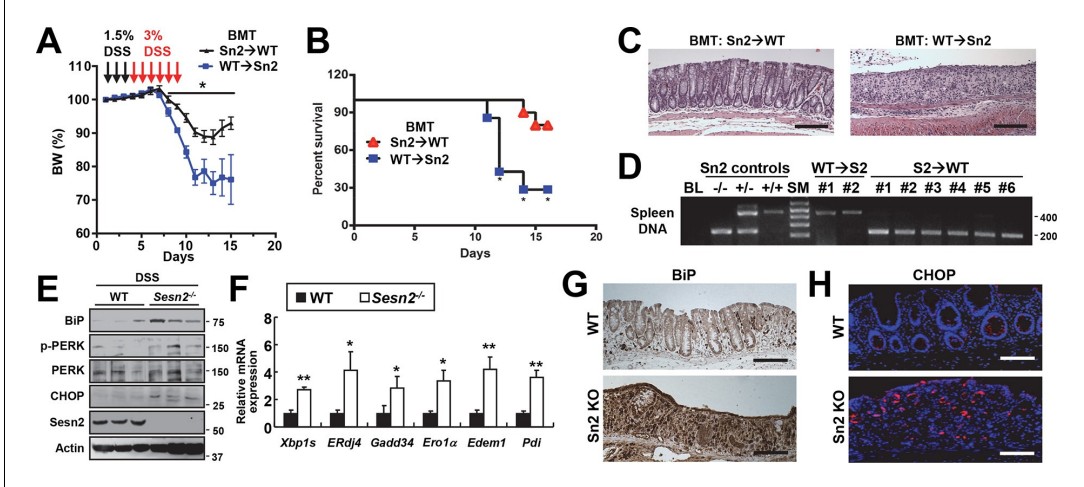

**Figure 2.** Sestrin2 prevents colitis-associated ER stress in colonic epithelia. (A-D) Sestrin2 expression in the extra-hematopoietic compartment is critical for the resolution of DSS-induced colitis. 3-month-old WT and *Sesn2*[-/-] mice (n=7 each) were subjected to lethal irradiation and injected with bone marrow cells from age-matched *Sesn2*[-/-] (Sn2→WT) and WT (WT→Sn2) mice, respectively. After 1 month, mice were subjected to DSS administration as indicated in the panel A. Body weight was measured over 15 days (A). Data are shown as mean ± s.e.m. Percent survival was calculated for each day (B). The colons of surviving mice at the final day of experiment were isolated and fixed for H&E staining (C). The spleens of surviving mice were genotyped for WT (upper band, ~450bp) and *Sesn2*-KO (lower band, ~200bp) alleles (D). (E-H) Loss of *Sesn2* aggravates colitis-induced ER stress in colon. Expression or phosphorylation of ER stress signaling markers were analyzed from indicated mice (described in Figure 1D-M) through immunoblotting (E), real-time PCR (F) or immunohistochemistry (G,H). Data are shown as mean ± s.e.m. *p<0.05, **p<0.01. *P* values are from Student's t-test. Scale bars, 200 μm (black), 100 μm (white). Molecular weight markers are indicated in bp (D) or kDa (E).

The following figure supplements are available for figure 2:

**Figure supplement 1.** Induction of ER stress and Sestrin2 upon DSS treatment.
**Figure supplement 2.** Increased ER stress in colon epithelia of *Sesn2*[-/-] and *Sesn2*[-/-]/*Sesn3*[-/-] mice during DSS-induced colon injury.
**Figure supplement 3.** Uncropped images of blots.

irradiated WT mice (Sn2→WT). Both groups of chimeric mice were subjected to DSS administration (*Figure 2A*). Interestingly, Sn2→WT mice were similar to WT mice and able to recover from DSS-induced injury (*Figure 2A,B*). However, more than 70% of WT→Sn2 mice were dead at 6 days following DSS treatment (*Figure 2B*). WT→Sn2 mice, but not Sn2→WT mice, also experienced dramatic weight loss during the recovery phase following DSS treatment (*Figure 2A*). Histological examination of the surviving mice revealed that, although the epithelial structure of Sn2→WT mouse colon was nearly completely restored at 6 days after the DSS treatment, WT→Sn2 colon epithelium was degenerated and marked by complete loss of epithelial cells and a robust increase in infiltrating immune cells (*Figure 2C*). Genotyping of spleen tissues in surviving WT→Sn2 and Sn2→WT mice confirmed that the hematopoietic compartment of recipient mice had been completely substituted with bone marrow cells of the donor (*Figure 2D*). These results indicate that the extra-hematopoietic presence of Sestrin2, such as in epithelial cells, is critical for the maintenance of intestinal homeostasis during colitis.

## Sestrin2 deficiency impairs recovery from DSS-induced ER stress

Sestrin2 protects cells and tissues from ER stress and its pathological sequelae such as metabolic abnormalities, tissue inflammation and apoptotic cell death (*Park et al, 2014*). DSS treatment was recently found to induce ER stress in colon epithelia (*Cao et al, 2013*), and consistent with this report, we observed that DSS treatment can induce prolonged ER stress signaling that is associated with modestly elevated Sestrin2 expression (*Figure 2—figure supplement 1*). Therefore, it is possible that Sestrin2 protects colon epithelium by allowing colonocytes to cope with DSS-induced ER

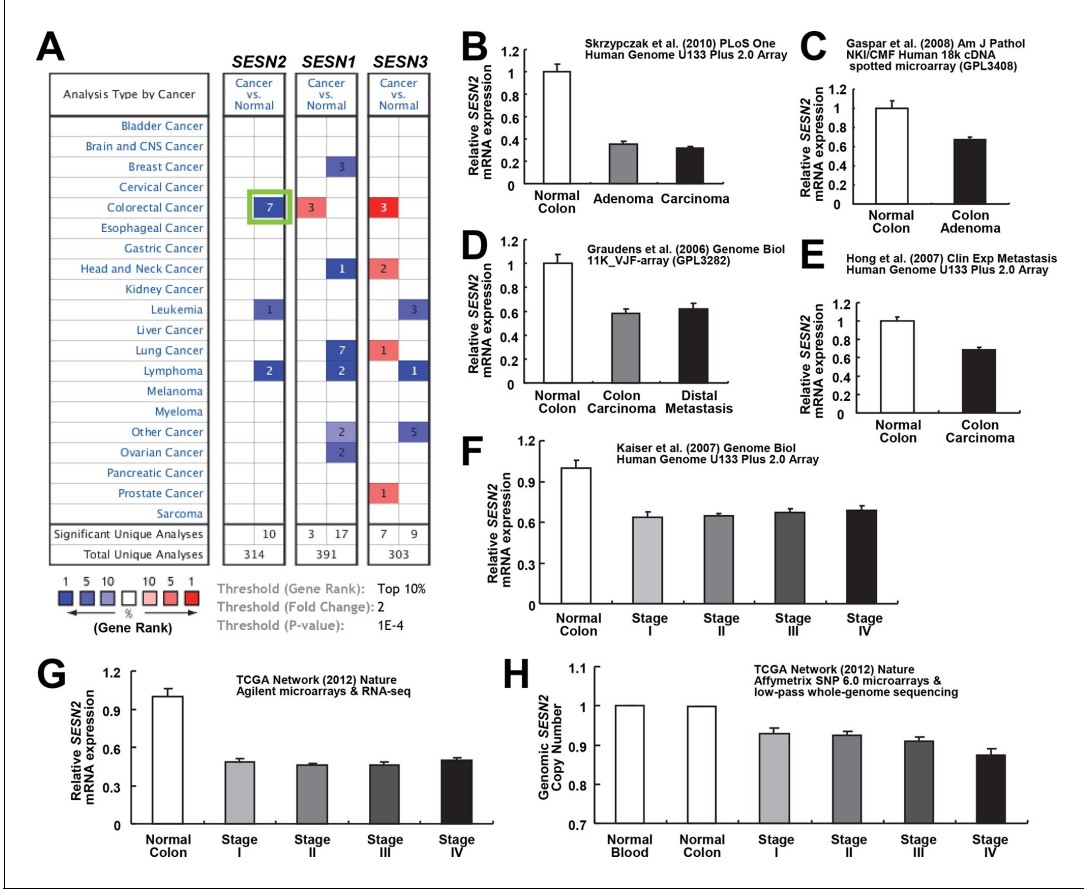

**Figure 3.** Downregulation of *SESN2* in human colon cancer tissues. (**A**) Oncomine analysis of Sestrin-family gene expression in normal and cancer tissues of different types. Gene summary views for *SESN2, SESN1* and *SESN3* genes are shown. Cell color is determined by the best gene rank percentile for the analyses within the cell, as described below the table. Thresholds for gene rank, fold change and *P* value are also described below the table. Reduction of *SESN2* in colorectal cancer tissue (highlighted in the green box) was one of the most significant alterations. (**B-G**) human *SESN2* mRNA expression in normal colon and colon cancer tissues, derived from six independent studies (**B-G**; total n=40, 78, 58, 82, 80 and 165, respectively) conducted in different platforms (*Gaspar et al, 2008*; *Graudens et al, 2006*; *Hong et al, 2010*; *Kaiser et al, 2007*; *Cancer Genome Atlas Network, 2012*; *Skrzypczak et al, 2010*). (**H**) DNA copy number analysis of human *SESN2* gene in normal blood, normal colon and colon cancer tissues (total n=975), derived from TCGA dataset (*Cancer Genome Atlas Network, 2012*). Colon cancer staging in F, G and H is according to the TNM staging system from the American Joint Committee on Cancer (AJCC). All data are shown as the mean ± s.e.m. *P* values between normal and cancer tissues, calculated from Student's t-test, are all below $10^{-4}$ (**B-H**). For TCGA dataset, *P* values between normal and cancer tissues are $1.6 \times 10^{-30}$ (**G**) and $3.7 \times 10^{-58}$ (**H**).

The following figure supplement is available for figure 3:

**Figure supplement 1.** Expression of cell-type specific markers in human colon cancer tissues.

stress insults, and thereby promoting colonic recovery after DSS injury. To test this idea, we analyzed the level of ER stress in WT and *Sesn2$^{-/-}$* colon epithelium recovering from DSS injury. Immunoblot analyses of colon tissue showed that *Sesn2$^{-/-}$* and *Sesn2$^{-/-}$/Sesn3$^{-/-}$* mice have elevated phosphorylation of pancreatic ER kinase (PERK) and increased expression of BiP and CHOP (*Figure 2E* and *Figure 2—figure supplement 2A,B*), when compared to WT mice. mRNA expression analysis for ER stress target genes, including spliced XBP1 (XBP1s) and BiP cofactor ERdj4, were also robustly upregulated in the colon of *Sesn2$^{-/-}$* mice (*Figure 2F* and *Figure 2—figure supplement 2C*). Immunostaining of BiP and CHOP also confirmed the presence of prominent ER stress in the damaged colon epithelium of *Sesn2$^{-/-}$* mice (*Figure 2G,H* and *Figure 2—figure supplement 2D,E*). These results collectively indicate that endogenous Sestrin2 is critical for the suppression of colon ER stress after DSS insults.

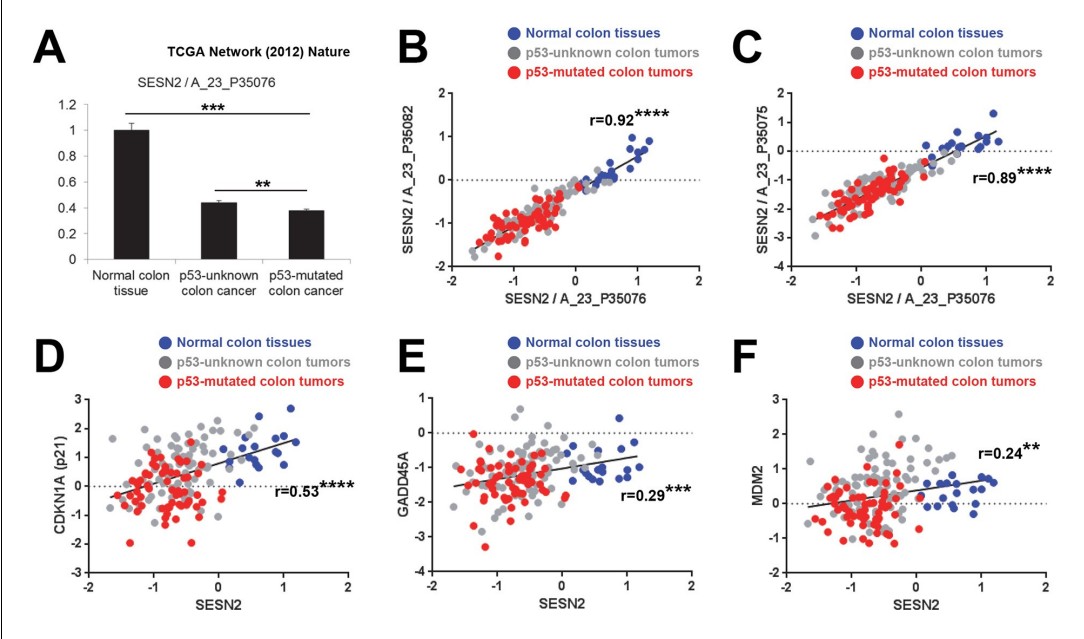

**Figure 4.** Correlation between p53 status and *SESN2* expression in human colon cancer. TCGA colon/colon cancer dataset (total n=165) was partitioned according to the p53 status (*Cancer Genome Atlas Network, 2012*). Normal colon tissues do not reveal any *TP53* mutation (n=19 in the dataset), while ~43% of tumor samples in the gene expression dataset (n=63 out of total 146 tumor samples), designated as 'p53-mutated', identified missense or nonsense point mutations in the *TP53* coding region. Other tumor samples (n=83) are designated as 'p53-unknown'. (A) *SESN2* expression was analyzed in indicated tissues. Data are shown as the mean ± s.e.m. **p<0.01,***p<0.001. *P* values were calculated from Student's t-test. (B,C) Expression of *SESN2* was analyzed in three different probes, and their correlations were visualized by a scatter plot of individual patient tissue samples. Trend lines were approximated through linear regression. Y axis is in a log scale. Normal colon samples are in blue, 'p53-unknown' tumor samples are in gray, and 'p53-mutated' tumor samples are in red. Pearson's correlation coefficients (r) with *P* values were calculated and presented. ****p<0.0001. (D-F) Expression of *SESN2*, *CDKN1A*, *GADD45A* and *MDM2* was analyzed and their correlations were analyzed as described above for B and C. **p<0.01, ***p<0.001, ****p<0.0001.

The following figure supplement is available for figure 4:

**Figure supplement 1.** Correlation between p53 status and expression of *CDKN1A, GADD45A, MDM2, BAX, PUMA (BBC3), p53AIP1 (TP53AIP1), TSC2, AMPKβ (PRKAB1, PRKAB2)* and *PTEN* in human colon cancer.

## Downregulation of *SESN2* in human cancer tissues

Our results show that Sestrin2 expression is induced during UC as a protective mechanism against colonic ER stress and epithelial degeneration. Since pre-existing colitis (*Terzic et al, 2010*) or tumor-elicited inflammation (*Grivennikov et al, 2012*) is important for the progression of colon cancer, it is possible that Sestrin2 expression is lost or downregulated during colon carcinogenesis. Thus, we examined the Oncomine database, which contains diverse large-scale genomic and transcriptomic analyses of normal and tumor tissues (*Rhodes et al, 2007*), to determine if there is a differential expression of Sestrins between normal and tumor tissues. Surprisingly, in virtually all of the colon cancer transcriptome studies available within the database, which were conducted in diverse platforms using different patient tissues, human Sestrin2 (*SESN2*) mRNA expression was strongly downregulated in colon adenocarcinoma tissues when compared to normal colon controls (*Figure 3A–G*) (*Gaspar et al, 2008*; *Graudens et al, 2006*; *Hong et al, 2010*; *Kaiser et al, 2007*; *Cancer Genome Atlas Network, 2012*; *Skrzypczak et al, 2010*). The magnitude of *SESN2* suppression in tumors is often among the top 1–5% of all suppressed genes (*Figure 3A*). Considering that these data are collected from a variety of different human samples, the extent of the difference was very strong (all studies indicate p<10⁻⁴ or much lower). In contrast, other major cell type markers, such as *Villin (VIL1*, enterocytes), *DLL1* (progenitor cells), *F4/80 (EMR1*, macrophages), *Gr-1 (LY6G5B*, leukocytes), or *β-catenin (CTNNB*, colon epithelia), did not show such strong suppression (*Figure 3—figure supplement 1A–E*), while the stem cell marker *LGR5* was rather upregulated in tumor samples

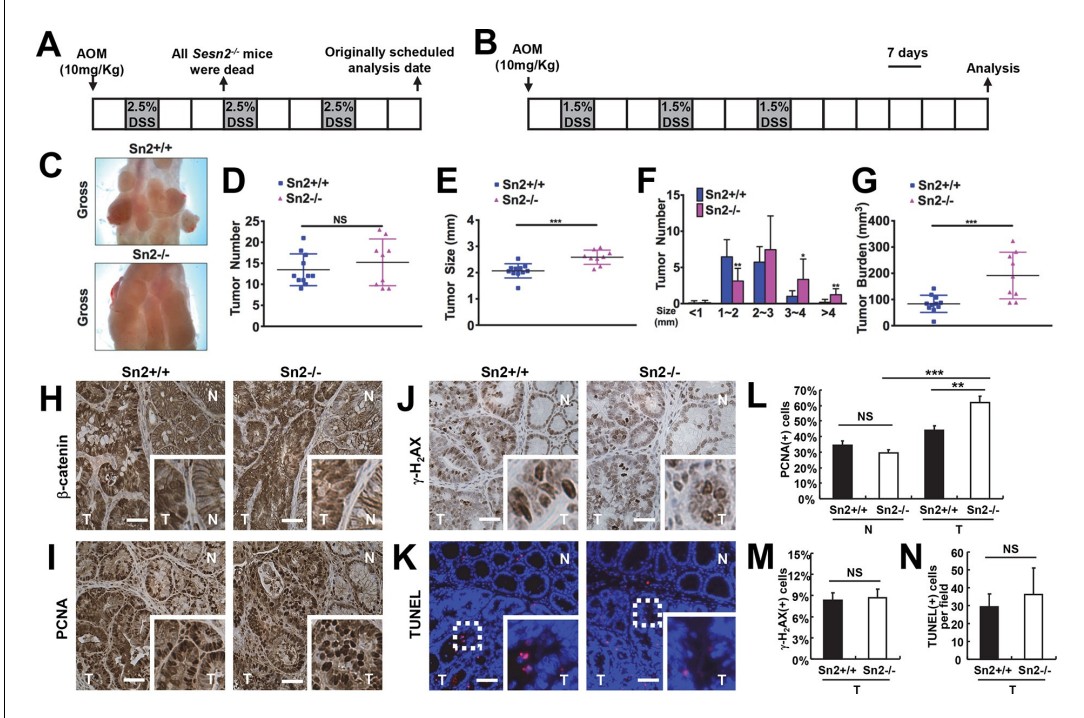

**Figure 5.** Loss of *Sesn2* promotes colon tumor growth in mice. (A) 2-month-old WT and *Sesn2⁻/⁻* mice were exposed to a standard protocol of azoxymethane (AOM)-dextran sulfate sodium (DSS)-induced colon carcinogenesis as outlined in this figure panel. However, none of the *Sesn2⁻/⁻* mice (n=7) survived after the first round of administration of DSS , suggesting that they are hypersensitive to DSS-induced colitis. (B) WT and *Sesn2⁻/⁻* mice (n=11 each) were subjected to a modified protocol of AOM-DSS-induced colon cancer. By reducing the dose of DSS to 1.5%, we were able to keep a substantial number of *Sesn2⁻/⁻* mice (n=9) alive until the experimental endpoint. Tumor incubation period was extended to 100 days to compensate for the lower dose of DSS treatment. (C-G) After completion of the experiment, colons were examined under a dissection microscope (C), and tumor number (D), average tumor size (E), size of individual tumors (F) and total tumor burden (G) were analyzed and presented as means from each mouse and as mean ± s.d. of the whole groups. (H-N) Sestrin2-deficient tumors exhibit increased proliferation. Colon tumor (T) and normal colon (N) tissues of indicated mice were subjected to immunohistochemistry of β-catenin (H), PCNA (I) and γ-H₂AX (J) or TUNEL staining (K). PCNA- (L), γ-H₂AX- (M) and TUNEL-positive (N) cells from indicated tissues were quantified and presented as mean ± s.e.m. NS, not significant; *p<0.05, **p<0.01, ***p<0.001. *P* values are from Student's t-test. Scale bars, 100 μm.

(*Figure 3—figure supplement 1A–E*). These results indicate that the downregulation of *SESN2* mRNA in colon cancer is specific and not an indirect consequence of different compositions of cell subtypes between normal and tumor tissues.

*SESN2* genome copy is also significantly reduced in colon cancer, and decreases further as the colon cancer progresses (*Figure 3H*). However, the extent of copy number loss was very small (~10%), suggesting that transcriptional downregulation, rather than the loss of genomic information, is the major mechanism of *SESN2* inhibition during colon cancer progression.

### *SESN2* expression exhibits strong correlation with p53 status

*SESN2* is a transcriptional target of tumor suppressor p53 (*Budanov et al, 2002*), which is one of the most frequently mutated genes in colon cancer (*Fearon, 2011*). To test whether p53 mutation plays any role in regulating *SESN2* expression during colon carcinogenesis, we analyzed the cancer genome atlas (TCGA) dataset (*Cancer Genome Atlas Network, 2012*) by partitioning tumors based on p53 status. TCGA dataset has comprehensive information regarding the genomic status of each tumor, determined by whole genome/exome sequencing. From this database, we were able to classify all the colon tumor samples into the two groups. One group, designated as 'p53-mutated', has one or more missense or nonsense coding sequence mutations in the *TP53* gene. The second group, designated as 'p53-unknown', is classified as such because it does not reveal any coding sequence mutations in the *TP53* gene; however, it is still possible that these tumors contain *TP53* mutations in

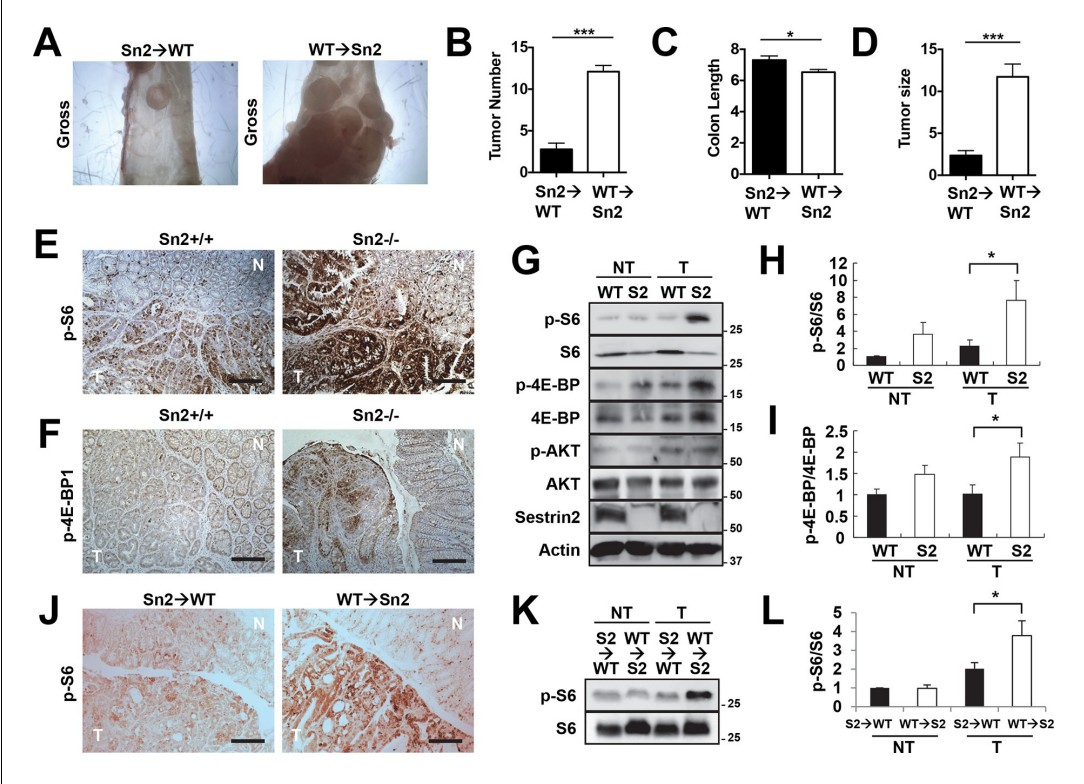

**Figure 6.** Sestrin2 loss increases mTORC1 signaling in colon cancer. (A-D) 3-month-old WT and *Sesn2*[-/-] mice (n=11 each) were subjected to lethal irradiation and injected with bone marrow cells from age-matched *Sesn2*[-/-] (Sn2→WT) and WT (WT→Sn2) mice, respectively. After 1 month, mice were subjected to AOM/DSS administration as indicated in *Figure 5B*. After completion of the experiment, colons were examined under a dissection microscope (A), and tumor number (B), colon length (C) and tumor size (D) were analyzed and presented as mean ± s.e.m. (E, F and J) Colon tumor (T) and normal colon (N) tissues of indicated mice were subjected to immunohistochemistry of phospho-Ser235/236-S6 (E and J) or phospho-Thr37/46-4E-BP (F). (G-I, K and L) Colon tumor (T) and normal colon (NT) tissues of WT and *Sesn2*[-/-] (S2) mice (G-I), as well as WT→Sn2 and Sn2→WT mice (K and L), were subjected to immunoblotting of indicated mTORC1 and mTORC2 signaling markers (G and K). Relative band intensities were quantified through densitometry and presented as mean ± s.e.m (H, I and L; n=6 in each group). *p<0.05, ***p<0.001. *P* values are from Student's t-test. Scale bars, 200 μm. Molecular weight markers are indicated in kDa.

The following figure supplements are available for figure 6:

**Figure supplement 1.** Efficiency of bone marrow transplantation.

**Figure supplement 2.** Additional immunohistochemistry results for *Figure 6E* (A), *Figure 6F* (B) and *Figure 6J* (C).

**Figure supplement 3.** Additional immunoblotting results for *Figure 6G* (A) and *Figure 6K* (B).

**Figure supplement 4.** Uncropped images of blots.

essential non-coding regions (e.g. promoters, enhancers or introns) or other genomic or epigenetic alterations that can lead to functional p53 inactivation (e.g. MDM2 overexpression). Expressions of *SESN2* and other known p53 target genes were then analyzed in three different groups: normal colons, 'p53-unknown' tumors and 'p53-mutated' tumors (*Figure 4* and *Figure 4—figure supplement 1*).

The analyses demonstrated that *SESN2* expression is significantly reduced in 'p53-mutated' tumors when compared to 'p53-unknown' tumors (*Figure 4A*), suggesting a role of p53 in controlling *SESN2* expression. This reduction was consistently observed in three independent *SESN2* probes (*Figure 4A–C*). Strikingly, there were almost no overlaps of *SESN2* expression levels between the normal colon and 'p53-mutated' tumor groups (*Figure 4B,C*), while *SESN2* levels in 'p53-

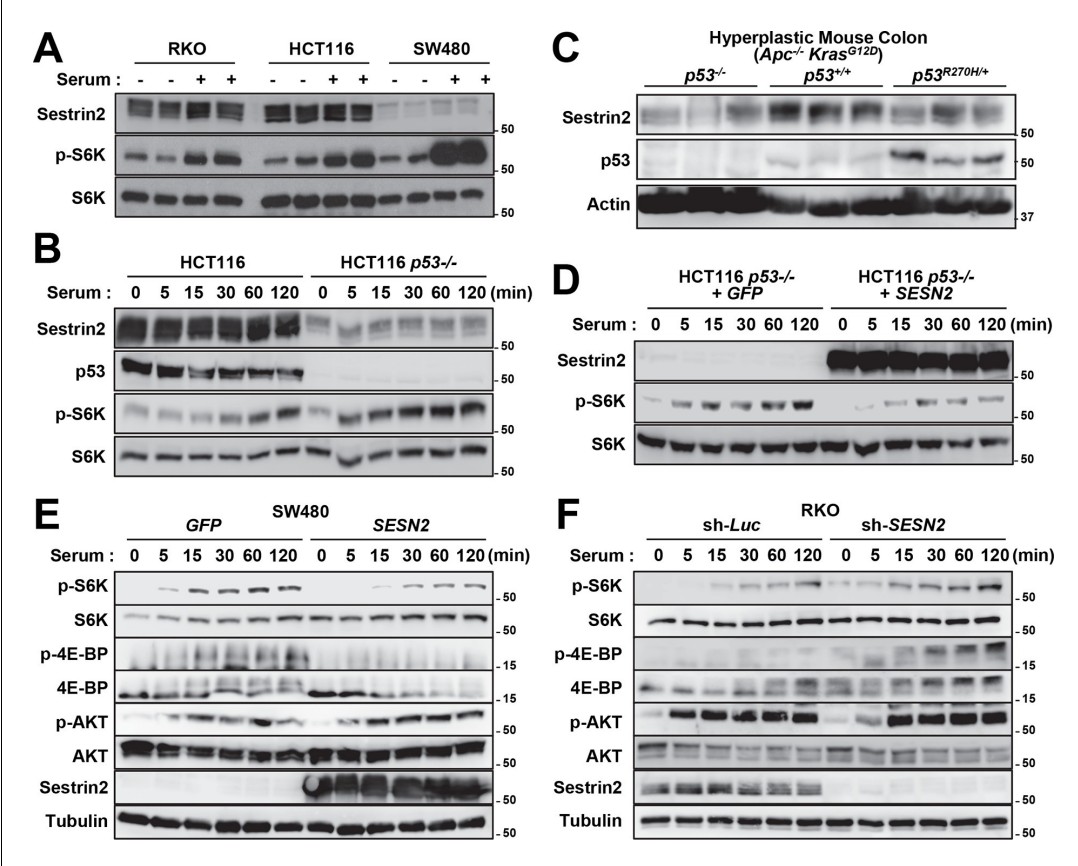

**Figure 7.** p53 controls mTORC1 signaling through Sestrin2 in colon cancer cells. (**A-F**) Whole cell or tissue lysates from the following experiments were subjected to immunoblotting of indicated proteins. (**A**) Human colon cancer cell lines (RKO, HCT116 and SW480) were serum-starved for 24 hr and then treated with 10% FBS (serum) for 2 hr. (**B**) p53-knockout (HCT116 $p53^{-/-}$) and control HCT116 cells were serum-starved for 24 hr and then treated with 10% FBS for indicated time. (**C**) $CDX2P-CreER^{T2}Apc^{flox/flox} Kras^{LSL-G12D/+} p53^{flox/flox}$ (left three lanes), $CDX2P-CreER^{T2}Apc^{flox/flox} Kras^{LSL-G12D/+} p53^{+/+}$ (centre three lanes), and $CDX2P-CreER^{T2}Apc^{flox/flox} Kras^{LSL-G12D/+} p53^{R270H/+}$ (right three lanes) mice (*Feng et al, 2013*) were daily injected with 100 mg/kg tamoxifen (i.p.) for 3 days, and dysplastic colon tissues were harvested after 8 days. (**D**) HCT116 $p53^{-/-}$ cells were infected with GFP- or Sestrin2-expressing lentiviruses. After 24 hr, cells were serum-starved for 24 hr and treated with 10% FBS for indicated time. (**E**) SW480 cells were infected with GFP- or Sestrin2-expressing lentiviruses. After 24 hr, cells were serum-starved for 24 hr and treated with 10% FBS for indicated time. (**F**) RKO cells, stably infected with lentiviruses expressing shRNA targeting luciferase (sh-*Luc*) or the *SESN2* (sh-*SESN2*) gene, were serum-starved for 24 hr and then treated with 10% FBS for indicated time. Molecular weight markers are indicated in kDa.

The following figure supplements are available for figure 7:

**Figure supplement 1.** Sestrin2 inhibits colon cancer cell growth through mTORC1 inhibition.

**Figure supplement 2.** Sestrin2-deficiency renders cancer cells less sensitive to chemotherapy.

**Figure supplement 3.** Uncropped images of blots.

unknown' tumors overlap with both groups (*Figure 4B,C*). Other known p53 target genes, such as *CDKN1A (p21)*, *GADD45A* and *MDM2* (*Riley et al, 2008*), did not show this strong correlation: for these genes, considerable overlaps were found between the normal colon and 'p53-mutated' tumor groups in individual samples (*Figure 4D–F*). Nevertheless, expression levels of these genes have a general positive correlation with expression of *SESN2* in individual samples (*Figure 4D–F*), suggesting that *SESN2*, *CDKN1A*, *GADD45A* and *MDM2* are all regulated through the same p53-dependent mechanism. Indeed, all of these genes are differentially expressed between 'p53-unknown' and 'p53-mutated' tumor groups (*Figure 4A* and *Figure 4—figure supplement 1A–C*).

Interestingly, some p53 target genes, such as apoptosis mediators *BAX*, *PUMA (BBC3)* and *p53AIP1 (TP53AIP1)* (*Riley et al, 2008*) or mTORC1 regulators *TSC2*, *AMPKβ (PRKAB1* and *PRKAB2)* and *PTEN* (*Feng et al, 2007*), did not show differential expression between the 'p53-unknown' and 'p53-mutated' tumor groups (*Figure 4—figure supplement 1D–J*), suggesting that the effects of p53 mutations on these genes are minimal in the pathological context of colon carcinogenesis.

These results collectively highlight *SESN2* as a clinically relevant target of p53 during colon carcinogenesis.

### *Sesn2* loss promotes colitis-induced colon cancer growth

To more precisely establish the cause-effect relationship between Sestrin2 and colon cancer development, *Sesn2⁻/⁻* mice were further assessed. The analysis of two-year-old WT and *Sesn2⁻/⁻* mice (n=6 each) did not reveal any noticeable colon tumors, suggesting that *Sesn2* may not be a classical tumor suppressor gene whose homozygous deletion is sufficient to induce spontaneous cancer development. However, given the strong correlation between *SESN2* expression and colon cancer tumorigenesis in humans (*Figure 3*), we reasoned that endogenous Sestrin2 may play a role in attenuating tumor development and progression. Thus, *Sesn2⁻/⁻* mice were subjected to an established protocol of colitis-associated colon cancer induction; in this model, colon carcinogenesis is initiated by azoxymethane (AOM) administration and promoted by repeated colon injury induced by 2.5% DSS. However, none of *Sesn2⁻/⁻* mice (n=7) survived the first round of colitis induction (*Figure 5A*), while WT mice were able to survive the entirety of the treatment. This is consistent with our findings that Sestrin2 has an important physiological role in maintaining epithelial integrity during colon injury (*Figures 1–2*).

To overcome the DSS-induced lethality of *Sesn2⁻/⁻* mice, a lower dose of DSS (1.5%) was administered to both control and experimental groups in the colitis-associated cancer model (*Figure 5B*). This modification enabled both WT and *Sesn2⁻/⁻* groups to survive through three inflammatory and recovery phases, and develop macroscopically visible colon tumors at 100 days after AOM injection (*Figure 5C*). There was no substantial difference in the number of tumors between WT and *Sesn2⁻/⁻* mice (*Figure 5D*). However, tumor size (*Figure 5E,F*) and burden (*Figure 5G*) were dramatically increased in *Sesn2⁻/⁻* mice, suggesting that Sestrin2 loss promoted tumor growth and progression in the colitis-associated cancer model. Tumors developed in both WT and *Sesn2⁻/⁻* mice displayed classical characteristics of colon adenomas, such as nuclear β-catenin staining (*Figure 5H*), increased cell proliferation (*Figure 5I*; PCNA staining), DNA damage (*Figure 5J*; γ-H2AX staining) and apoptotic cell death (*Figure 5K*; TUNEL staining). Quantification of PCNA-, γ-H2AX- and TUNEL-positive cells showed that tumor cell proliferation (*Figure 5L*) is significantly increased in tumors isolated from *Sesn2⁻/⁻* mice, while DNA damage (*Figure 5M*) and apoptosis (*Figure 5N*) were not significantly altered by the loss of Sestrin2.

### Sestrin2 suppresses tumor growth in the extra-hematopoietic compartment

Using the reciprocal bone marrow chimera experiments followed by the AOM-DSS treatments (*Figure 6—figure supplement 1*), we assessed whether the expression of Sestrin2 in the hematopoietic or extra-hematopoietic compartment is important for the tumor-suppressive role of Sestrin2 in colon cancer. Compared to the Sn2→WT mice, WT→Sn2 mice exhibited strongly exacerbated tumor growth phenotypes, which was evident in both tumor number and size (*Figure 6A–D*), indicating that the extra-hematopoietic expression of Sestrin2, such as in colon epithelia, is critical for colon tumor suppression.

### *Sesn2* loss induces mTORC1 hyperactivation

It is possible that epithelial Sestrin2 expression tissue-autonomously suppresses colon tumor growth. Indeed, Sestrin2 is a potent inhibitor of mTORC1 (*Budanov and Karin, 2008*; *Lee et al, 2013*), which is known to be critical for tumorigenic growth in colon cancer cells (*Faller et al, 2015*; *Hardiman et al, 2014*). Therefore, we analyzed mTORC1 downstream target proteins in colon cancer and normal colon tissues from WT and *Sesn2⁻/⁻* mice through immunohistochemistry. Phosphorylation of ribosomal protein S6, which is mediated by an mTORC1 target p70 S6 kinase (S6K), is dramatically increased in tumors isolated from *Sesn2⁻/⁻* mice (*Figure 6E* and *Figure 6—figure*

supplement 2A). Consistent with this observation, phosphorylation of eukaryotic translation initiation factor 4E-binding protein (4E-BP), an additional mTORC1 target, was also prominently increased in *Sesn2*[-/-] colon cancer tissues, when compared to adjacent normal colon tissues as well as to WT colon cancer tissues (*Figure 6F* and *Figure 6—figure supplement 2B*). These immunohistochemical observations were also confirmed by immunoblotting experiments; S6 and 4E-BP phosphorylation was prominently increased in colon tumors of *Sesn2*[-/-] mice, compared to those of WT mice (*Figure 6G–I* and *Figure 6—figure supplement 3A*). Tumors from WT→Sn2 mice, but not those from Sn2→WT mice, also exhibited hyperactive mTORC1 signaling, indicated by increased S6 phosphorylation (*Figure 6J–L* and *Figure 6—figure supplement 2C,3B*). However, AKT Ser473 phosphorylation, which is mediated by mTORC2, a complex distinct from mTORC1, was not significantly altered by Sestrin2 loss in colon cancer tissues (*Figure 6G*). Collectively, these data suggest that Sestrin2 loss selectively leads to mTORC1 hyperactivation in colon cancer tissues, which subsequently allowed for prominent tumor overgrowth.

## p53 regulates Sestrin2 expression in colon cancer cells

Unlike human colon cancer tissues (*Figures 3,4*), mouse colon cancers induced by the AOM-DSS treatment displayed Sestrin2 expression at a level comparable to that of normal colon tissues (*Figure 6G*). The expression of Sestrin2 in mouse colon tumor cells may provide an explanation for the relatively low mTORC1 activity in the tumors (*Figure 6E–L*) that do not progress to adenocarcinomas (*Rosenberg et al, 2009*). Interestingly, it has been formerly reported that most mouse colon tumors induced by the AOM-DSS treatment do not contain p53 mutations or misregulation (*Nambiar et al, 2004*; *Takahashi and Wakabayashi, 2004*), and this may provide an explanation of why Sestrin2 expression is sustained in this specific mouse model of colon cancer. Indeed, SW480 cells, a human colon cancer cell line that displays very low p53 activity, demonstrated a strong downregulation of Sestrin2 expression, while RKO and HCT116 cells, which have wild-type p53 activity, expressed a relatively high amount of Sestrin2 proteins (*Figure 7A*). The p53-deficient HCT116 cells exhibited dramatic downregulation of Sestrin2 expression when compared to the parental HCT116 cells (*Figure 7B*). Expression of Sestrin2 in hyperplastic mouse colon tissues, which have mutations in both *Apc* and *Kras* genes (*Feng et al, 2013*), was also reduced by the loss of p53 or a dominant-negative mutation of p53 (*Figure 7C*). Consistent with clinical data on human *SESN2* expression described above (*Figure 4*), these results further support the notion that p53 is critical for Sestrin2 expression in colon cancer cells.

## Sestrin2 suppresses clonogenic growth through mTORC1 inhibition

p53 inhibits mTORC1 signaling, and this regulation may be an important contributor to the tumor suppressive activity of p53 (*Agarwal et al, 2016*; *Feng and Levine, 2010*). To understand if Sestrin2, acting as a p53 target, has an essential role in regulating mTORC1 and colon cancer cell growth, Sestrin2 was overexpressed in p53-deficient HCT116 cells and SW480 cells, which have low *SESN2* expression and high mTORC1 signaling (*Figure 7A,B*). Restoration of Sestrin2 suppressed serum-induced phosphorylation of mTORC1 substrates, S6K and 4E-BP (*Figure 7D,E*), but not mTORC2 substrate AKT (*Figure 7E*). Sestrin2 silencing in RKO cells enhanced the mTORC1 signaling in both starved and serum-stimulated cells (*Figure 7F*), indicating that endogenous Sestrin2 indeed functions to inhibit mTORC1 signaling in human colon cancer cells.

As mTORC1 is known to control cell growth, we examined the effect of Sestrin2 on clonogenic growth of RKO and SW480 cells. A colony forming assay of Sestrin2-silenced RKO cells demonstrated an increase in clonogenic growth compared to the control cells (*Figure 7—figure supplement 1A,B*), suggesting that Sestrin2 is an inhibitor of cancer cell growth. To further investigate if this Sestrin2 activity is dependent on mTORC1 hyperactivation, we treated control and Sestrin2-silenced cells with rapamycin, an inhibitor of mTORC1. Rapamycin inhibited clonogenic growth in both control and Sestrin2-silenced cells, and interestingly, the growth difference between control and Sestrin2-silenced cells was diminished upon rapamycin treatment (*Figure 7—figure supplement 1A,B*). These results indicate that Sestrin2 attenuates cancer cell growth primarily through inhibition of a hyperactive mTORC1. On the other hand, expression of Sestrin2 in SW480 cells substantially inhibited clonogenic growth (*Figure 7—figure supplement 1C,D*), supporting the idea that Sestrin2 is critical for inhibition of colon cancer cell overgrowth.

## Sestrin2 loss confers chemotherapy resistance to colon cancer cells

As a stress-inducible cell growth regulator, Sestrin2 may be also important for the responsiveness of colon cancer cells to chemotherapeutic treatments. To examine this possibility, we treated control and Sestrin2-silenced RKO cells with two representative chemotherapeutic agents, 5-fluorouracil (5-FU) and irinotecan (CPT-11). Although control cell growth was strongly suppressed by 5-FU or CPT-11, Sestrin2 silencing rendered cells less susceptible to the chemotherapeutic drug treatments (*Figure 7—figure supplement 2A–C*), suggesting that Sestrin2 loss may confer chemotherapy resistance to colon cancer cells.

We also examined Sestrin2 expression and mTORC1 signaling in RKO cells when treated with 5-FU and CPT-11. After 5-FU or CPT-11 treatments, Sestrin2 expression was slightly elevated in WT cells (sh-Luc; *Figure 7—figure supplement 2D,E*). Interestingly, mTORC1 signaling, monitored by p-S6K and p-4E-BP1, was prominently upregulated in sh-*SESN2* cells after 5-FU treatment (*Figure 7—figure supplement 2D*), suggesting that Sestrin2 suppresses mTORC1 activation after 5-FU treatment. In contrast, CPT-11 treatment reduced mTORC1 signaling, and Sestrin2 silencing led to modest but persistent mTORC1 upregulation (*Figure 7—figure supplement 2E*). This mTORC1 upregulation as a result of Sestrin2 loss could have conferred chemoresistance to RKO cells against 5-FU and CPT-11.

## Discussion

The current study reveals how the stress-inducible protein Sestrin2 can coordinate ER stress and mTORC1 signaling pathways to maintain epithelial homeostasis and limit colitis and colon cancer development during colon injury (*Figure 7—figure supplement 2F*). Yet, it is still possible that the mechanisms underlying increased susceptibility to colitis and colon cancer are separate from each other. During colitis stress, Sestrin2 functions to suppress ER stress and to promote regeneration of colon epithelium (*Figures 1,2*). However, Sestrin2 expression is lost during human colon carcinogenesis (*Figure 3*) through inactivation of tumor suppressor p53 (*Figures 4,7*). Because expression of Sestrin2 is important for suppressing hyperactive mTORC1 signaling (*Figure 6*) and tumor outgrowth (*Figure 5*), loss of Sestrin2 expression in human colon cancer serves as a critical tumorigenic mechanism. Indeed, Sestrin2 negatively controlled cell growth in various human colon cancer cell lines, which was dependent on mTORC1 regulation (*Figure 7—figure supplement 1*). Furthermore, inactivation of Sestrin2 conferred chemoresistance to colon cancer cells (*Figure 7—figure supplement 2*), rendering them difficult to treat with conventional chemotherapeutic methods.

In addition to mTORC1 regulation, Sestrin2 is also known to reduce oxidative stress (*Budanov et al, 2004*) by functioning as an activator of anti-ROS transcription factor Nrf2 (*Bae et al, 2013*) or as an alkylhydroperoxidase (*Kim et al, 2015a*). Thus, it is possible that the loss of Sestrin2 can contribute to cellular accumulation of ROS, which can promote DNA damage and genomic mutations that facilitate tumor development (*Sablina et al, 2005*). However, analysis of γ-H2AX did not show a significant increase in DNA damage between colon tumors of WT and *Sesn2*$^{-/-}$ mice (*Figure 5J,M*). This result suggests that the mTORC1-regulatory function, rather than the ROS-inhibiting function, is the main contributor of Sestrin2's tumor suppressive activity in colon tissues. Nevertheless, it is still possible that Sestrin2 attenuates tumor growth, at least partially by inhibiting tumor-associated epithelial damage and inflammation (*Figures 1,2*), which are well-characterized promoters of colon cancer growth (*Grivennikov et al, 2012*; *Terzic et al, 2010*). It is also possible that Sestrin2 exerts tumor suppressive activity additionally through its apoptosis-inducing function, which was recently discovered (*Ding et al, 2015*).

The mTORC1-suppressing and ROS-reducing activities are shared between all members of the Sestrin family (Sestrin1-3) (*Lee et al, 2013*; *Nogueira et al, 2008*). However, only Sestrin2 was shown to be downregulated in human colon cancer tissues, and expression of Sestrin1 and Sestrin3 was unchanged or slightly upregulated in the colon cancer tissues. Nevertheless, Sestrin1 and Sestrin3 are strongly downregulated in several types of cancer tissues, such as lung cancers and lymphomas (*Figure 3A*), suggesting that they may be involved in anti-tumorigenic processes in tissues other than the colon. *Drosophila* Sestrin, which is the only Sestrin homologue expressed in *Drosophila*, was formerly shown to be a feedback inhibitor of mTORC1, which can suppress hyperplastic tissue growth provoked by oncogenic mTORC1 hyperactivation (*Lee et al, 2010*). As the current study demonstrates Sestrin2 to be an inhibitor of colon cancer growth, future studies on the role of

Sestrin1 and Sestrin3 in carcinogenic processes of other tissues may reveal a conserved tumor-suppressive role of Sestrin-family proteins.

## Materials and methods

### Antibodies and reagents

For immunoblotting, we obtained S6K (sc-230), PERK (sc-13073) and p-PERK (sc-32577) antibodies from Santa Cruz Biotechnology, Dallas, TX, human Sestrin2 (10795-1-AP) antibody from Proteintech, Chicago, IL, BiP (3177), CHOP (2895), p-Thr389 S6K (9234), p-Ser235/236 S6 (2211), S6 (2317), p-Thr37/46 4E-BP (2855), 4E-BP (9452), p-Ser473 AKT (9273) and AKT (4691) antibodies from Cell Signaling Technology, Danverse, MA, actin (JLA20) antibody from Developmental Studies Hybridoma Bank (DSHB, Iowa city, IA), and tubulin (T5168) antibody from Sigma, St. Louis, MO. Mouse Sestrin2 antibody was described (*Ro et al, 2014b*). For immunostaining, we obtained p53 (sc-6243), PCNA (sc-7907), $\beta$-catenin (sc-59737), CHOP (sc-575) from Santa Cruz Biotechnology, F4/80 (mf48000) from Invitrogen, Carlsbad, CA, BiP (3177), $\gamma$-H2AX (2577), p-Ser235/236 S6 (2211) and p-Thr37/46 4E-BP (2855) from Cell Signaling Technology. Azoxymethane (AOM), dextran sulfate sodium (DSS), rapamycin, 5-fluorouracil (5-FU) and irinotecan (CPT-11) were from Sigma.

### Cell culture

Human colon cancer cell lines, including RKO, SW480 and HCT116, were obtained from American Type Culture Collection (ATCC, Manassas, VA) and cultured in Dulbecco's modified Eagle's medium (DMEM, Invitrogen) containing 10% fetal bovine serum (FBS, Sigma), 50 U/ml penicillin and 50 mg/ml streptomycin. The cells were authenticated by Short Tandem Repeat (STR) profiling at ATCC, tested negative for mycoplasma infection, and subcultured for less than 6 months prior to initiation of the described experiments. *p53*-knockout HCT116 cells were obtained from Dr. Bert Vogelstein (Johns Hopkins University, Baltimore, MD). The p53 loss in this cell line was confirmed by western blot. All cultures were maintained in a 37°C incubator with 5% $CO_2$. The lentiviral constructs for Sestrin2 overexpression and silencing are formerly described (*Budanov and Karin, 2008*). Viruses were generated and amplified in the Vector Core facility at the University of Michigan (UM).

### Immunoblotting

Cells and tissues were lysed in RIPA buffer (50 mM Tris-Cl pH 7.4, 150 mM NaCl, 1% sodium deoxycholate, 1% NP-40; 0.1% SDS) or cell lysis buffer (20 mM Tris-Cl pH 7.5, 150 mM NaCl, 1 mM EDTA, 1 mM EGTA, 2.5 mM sodium pyrophosphate, 1 mM $\beta$-glycerophosphate, 1 mM $Na_3VO_4$, 1% Triton-X-100) containing protease inhibitor cocktail (Roche, Indianapolis, IA), and processed as formerly described (*Ro et al, 2014a*). Protein samples were boiled in SDS sample buffer for 5 min, separated by SDS-PAGE, transferred to PVDF membranes and probed with primary antibodies (1:200 for Santa Cruz antibodies, and 1:1000 for all other antibodies). After incubation with secondary antibodies conjugated with HRP (Bio-rad; 1:2000), chemiluminescence was detected using LAS4000 (GE, Fairfield, CT) systems or X-ray films. Immunoblot images were quantified by densitometry, and protein expressions were presented as relative band intensities. Uncropped images of immunoblots are provided in *Figure 2—figure supplement 3*, *Figure 6—figure supplement 4* and *Figure 7—figure supplement 3*.

### Quantitative reverse transcriptase-real time PCR

Total RNA was extracted from tissues or cells using Trizol reagent (Invitrogen), and cDNA was made using MMLV-RT (Promega, Madison, WI) and random hexamers (Invitrogen). Quantitative PCR was performed in a Real-Time PCR detection system (Applied Biosystems, Foster city, CA) with iQTM SYBR Green Supermix (Bio-rad, Hercules, CA) and relevant primers. Relative mRNA expression was calculated from the comparative threshold cycle (Ct) values relative to $\beta$-Actin. Primers for *Sestrins*, inflammation markers (*Tnfa*, *Il6*, *Il1b* and *Il10*), ER stress markers (*Xbp1s*, *ERdj4*, *Gadd34*, *Ero1α*, *Edem1*, and *Pdi*) and $\beta$-Actin were formerly described (*Park et al, 2010*; *Park et al, 2014*; *Ro et al, 2014a*).

## Database analyses

mRNA expression data and genome copy data from various studies were retrieved from Oncomine database (*Rhodes et al, 2007*). Exome sequencing information regarding the status of p53 was retrieved from Supplmentary Table 2 of TCGA colon cancer paper (*Cancer Genome Atlas Network, 2012*). TCGA data, retrieved from Oncomine, were manually partitioned into 'p53-mutated' and 'p53-unknown' groups according to the p53 gene status. When multiple probes for a same gene were found from the database, probes whose average values between normal and cancer groups (or between normal and 'p53-mutated' groups) are close to zero in log scale were selected for further analysis. Bar graphs were plotted in a linear scale and control values were normalized to one. Scatter plots were presented in a logarithmic scale using raw data. Correlation and linear regression analyses were performed in Graphpad Prism 6.

## Animal experiments

WT, $Sesn2^{-/-}$ and $Sesn2^{-/-}/Sesn3^{-/-}$ mice (*Lee et al, 2012*) and $CDX2P\text{-}CreER^{T2}Apc^{flox/flox}$ mice (*Feng et al, 2013*) were used for this study. These mice are on a C57BL/6 background. Mice were maintained in filter-topped cages and given free access to autoclaved regular chow diet at the UM according to the NIH and institutional guidelines. All animal studies were ethically approved (protocol approval numbers: PRO00005712 and PRO00004019) and overseen by the University Committee on Use and Care of Animals (UCUCA) at the UM.

For colitis induction, mice received water with 3% DSS for 6–7 days (inflammatory phase). Then the mice were placed on regular drinking water for 5–7 days (recovery phase) as formerly described (*Xue et al, 2013*). For tumor induction, mice were treated with AOM (10 mg/kg body weight). At 5 days following the AOM injection, mice received water with 1.5% DSS for 7 days (inflammatory phase). Then, the mice were placed on regular drinking water for 14 days (recovery phase). The mice were subjected to two more inflammatory and recovery cycles for tumor induction as in *Figure 5A, B*. Reciprocal bone marrow chimera experiments were done as described in our recent paper (*Anderson et al, 2013*), and the mice were subjected to DSS or AOM/DSS treatment at 1 month after the bone marrow transplantation to allow for complete substitution of the hematopoietic compartment. The mice in the same experiments are from an age-matched, co-housed cohort, and animal numbers were determined according to our previous studies (*Anderson et al, 2013*; *Xue et al, 2013*).

## Colon tumor phenotyping

A dissecting microscope (4x magnification) was used to assess the tumor number and size. Tumor size was defined as the mean of the two largest diameters measured with digital calipers. Tumor volume was derived from tumor size. Consistent with their histological appearance, a spherical shape was assumed for colon polyps, thus tumor volume = $4/3\pi r^3$, where r = radius. Tumor burden/load is defined as the total polyp volume per animal, which is the product of polyp number and polyp volume.

## Histology

Colons were removed, flushed with PBS, fixed in 4% paraformaldehyde at 4°C overnight and paraffin-embedded for histological analyses. Antigen retrieval was performed in 10 mM sodium citrate at 95°C for 15 min. For immunostaining of PCNA, $\beta$-catenin, F4/80, BiP, $\gamma$-H2AX, p-Ser235/236 S6 and p-Thr37/46 4E-BP, colon sections were incubated with corresponding primary antibodies (1:50, 1:50, 1:100, 1:200, 1:50, 1:200 and 1:100, respectively), followed by incubation with biotin-conjugated secondary antibodies (1:200) and streptavidin-HRP (1:300). The HRP activity was visualized with diaminobenzidine staining. Haematoxylin counterstaining was applied to visualize nuclei. For immunostaining of CHOP, colon sections were incubated with primary antibody (1:50), followed by incubation with Alexa 594-conjugated secondary antibody. DAPI counterstaining was applied to visualize nuclei. TdT-mediated dUTP nick end labeling (TUNEL) assay was performed using In Situ Cell Death Detection Kit TMR-Red (1215792910, Roche). The samples were analysed under an epi-fluorescence-equipped light microscope (Meiji MT6300).

## Acknowledgements

We thank Drs. M Karin (UCSD), AV Budanov (VCU), E Fearon, RA Miller, D Lombard, S Pletcher (UM), and Santa Cruz Biotech Inc. for sharing reagents and access to lab equipment. We also thank Jeongsoon Park for practical advice regarding clonogenic growth assay. This work was supported by grants from the Ellison Medical Foundation (AG-NS-0932-12, to JHL), Crohn's Colitis Foundation of America (276556, to XX), American Gastroenterological Association (to XX) and NIH (5T32GM008322 to A.H., CA148828 and DK095201 to YMS, DK102850 to JHL, and P30-AG024824, P30-AG013283, P30-DK034933, P30-DK089503 and P30-CA046592).

## Additional information

### Funding

| Funder | Grant reference number | Author |
|---|---|---|
| National Institutes of Health | CA148828, DK095201, DK102850, AG024824, AG013283, DK034933, DK089503, CA046592, GM008322 | Allison Ho<br>Yatrik M Shah<br>Jun Hee Lee |
| Lawrence Ellison Foundation | AG-NS-0932-12 | Jun Hee Lee |
| Crohn's and Colitis Foundation of America | 276556 | Xiang Xue |
| American Gastroenterological Association | | Xiang Xue |

The funders had no role in study design, data collection and interpretation, or the decision to submit the work for publication.

### Author contributions

S-HR, Performed biochemical, histological and cell biological analyses, Acquisition of data, Analysis and interpretation of data, Drafting or revising the article; XX, Performed animal experiments and analyses including DSS and AOM/DSS procedures, Acquisition of data, Analysis and interpretation of data; SKR, Assisted with animal procedures and analyses, Acquisition of data, Analysis and interpretation of data; C-SC, SN, H-WP, Assisted with cell culture experiments and biochemical analyses, Acquisition of data, Analysis and interpretation of data; IJ, Assisted with histological procedures and analyses, Acquisition of data, Analysis and interpretation of data; IAS, Assisted with cell culture experiments and biochemical analyses, Acquisition of data, Analysis and interpretation of data, Drafting or revising the article; AH, Assisted with histological procedures and analyses, Acquisition of data, Analysis and interpretation of data, Drafting or revising the article; YMS, JHL, Conception and design, Analysis and interpretation of data, Drafting or revising the article

### Author ORCIDs

Jun Hee Lee, http://orcid.org/0000-0002-2200-6011

### Ethics

Animal experimentation: All animal studies were ethically approved (protocol approval numbers: PRO00005712 and PRO00004019) and overseen by the University Committee on Use and Care of Animals (UCUCA) at the University of Michigan.

## Additional files

### Major datasets

The following previously published datasets were used:

| | Database, license, and accessibility |
|---|---|

| Author(s) | Year | Dataset title | Dataset URL | information |
|---|---|---|---|---|
| | 2007 | Oncomine 3.0: genes, pathways, and networks in a collection of 18,000 cancer gene expression profiles | https://www.oncomine.com | PMID: 17356713 |
| | 2008 | Data from: Cross-species comparison of human and mouse intestinal polyps reveals conserved mechanisms in adenomatous polyposis coli (APC)-driven tumorigenesis | https://www.oncomine.com | PMID: 18403596 |
| | 2006 | Data from: Deciphering cellular states of innate tumor drug responses | https://www.oncomine.com | PMID: 16542501 |
| | 2010 | Data from: A 'metastasis-prone' signature for early-stage mismatch-repair proficient sporadic colorectal cancer patients and its implications for possible therapeutics | https://www.oncomine.com | PMID: 20143136 |
| | 2007 | Data from: Transcriptional recapitulation and subversion of embryonic colon development by mouse colon tumor models and human colon cancer | https://www.oncomine.com | PMID: 17615082 |
| | 2012 | Data from: Comprehensive molecular characterization of human colon and rectal cancer | https://www.oncomine.com | PMID: 22810696 |
| | 2010 | Data from: Modeling oncogenic signaling in colon tumors by multidirectional analyses of microarray data directed for maximization of analytical reliability | https://www.oncomine.com | PMID: 20957034 |
| | 2012 | Data from: Comprehensive molecular characterization of human colon and rectal cancer | http://www.ncbi.nlm.nih.gov/pmc/articles/PMC3401966/bin/NIHMS379461-supplement-13.xlsx | PMID: 22810696 |

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
