## [Decision Letter]

Thank you for submitting your work entitled "Tumor suppressive role of Sestrin2 during colitis and colon carcinogenesis" for consideration by *eLife*. Your article has been favorably evaluated by Tony Hunter (Senior editor) and three reviewers, one of whom is a member of our Board of Reviewing Editors.

The reviewers have discussed the reviews with one another and the Reviewing Editor has drafted this decision to help you prepare a revised submission.

The current paper by Ro et al. demonstrates a tumor suppressive role for the SESN2 gene in colorectal cancer. Using a plethora of complementary approaches, the authors propose that loss of SESN2 is a key event during the development of colorectal cancer. The paper includes a number of mechanistic investigations depicting a tumor suppressive pathway whereby stress induces p53, which then induces SESN2, which then represses mTOR activity to suppress tumor growth.

The reviewers discussed the manuscript and decided to encourage resubmission of a revised manuscript addressing several major concerns:

1) Figure 1: It is not clear where these samples were obtained from and how they were analyzed. What was used as a control? Although there was a trend for mRNA increase of Sesn2 and Sesn3 in patients (n=10), the error bar is pretty big, which makes the data weak. If the authors really want to make this point, more patients samples are needed.

2) Figure 1. According to the working hypothesis, Sesn2 protects from ER stress and inflammation. Nevertheless, there is no significant difference between 2 mouse strains in body weight during first 7 days of treatment indicating that Sesn2 is not directly involved in control of inflammation and the observed phenotype might be a residual effect of Sesn2 inactivation. The effects of Sesn2 have to be assessed immediately after DSS treatment, and the impact of the DSS treatment on tissue morphology, regulation of Sesn2 protein expression, mTORC1 activity, ER stress levels and the inflammation should be addressed.

3) The mice used in Figure 1 and Figure 2 are different ages and are not comparable. It is not clear why the authors used 6-month old mice in Figure 1 rather than the younger mice as in Figure 2. Please clarify.

4) In Figure 2/F only DSS-treated mice are analyzed. To prove their claim the authors should compare the levels of ER stress in untreated and DSS-treated mice analyzing Sesn2 expression and the expression of the ER markers. Moreover, the protein extracts were not well normalized in Figure 2 and the actin marker is overexposed. Also the role of Sesn2 and ER stress in regulation of cell death is not addressed directly and not quantitated.

5) Figure 3: The subtle difference in mRNA levels between normal tissue and in tumors can be explained by compositions of different cell types in normal and tumor samples. Surprisingly, there is no significant downregulation of Sesn2 expression during cancer progression. Did the authors carry out an analysis of Sesn2 locus to find if Sesn2 locus has undergone LOH in tumors? Also related to this figure, unlike TP53, which is mutated in about half of all colorectal cancers, SESN2 is rarely mutated. The authors mined several public datasets demonstrating that SESN2 expression is lower in tumors relative to normal tissues. Is this lower expression in human tumors due to mutation or loss of p53? How is SESN2 expression between tumors expressing wild type versus mutant p53? Reanalysis of the various datasets in Figure 3 partitioning tumors based on p53 status would be a significant addition to the paper and overall model. Since the p53 field is heavily debating which of tis transcriptional targets carries tumor suppressive activity, this would be a good contribution to the field.

6) The experiments on cancer are not compatible with the experiments on colitis as different conditions were used for these experiments. What is the mouse age? Does Sesn2 protects from colitis, ER stress and cell death in the mice treated with 1.5% DSS in the colitis study. It is also not clear how tumor sizes and tumor load were analyzed.

7) In Figure 5 there is no evidence that Sesn2 expression and mTORC1 activity is somehow affected in tumor samples in control mice. Also the data on 4E-BP1 staining look odd because several differently phosphorylated forms of 4E-BP1 usually appear when cells are stained with 4E-BP1 antibody. Moreover, to analyze mTORC1 activity S6K phosphorylation data should be added to the experiment and several samples should be presented.

8) Figure 6: It was recently published by Agarwal et al., Mol. Can. Res 2015 that p53 inactivation in colon cancer cells led to strong suppression of Sesn2 expression as well as expression of some other p53-dependent mTORC1 regulators such as TSC2. Moreover, the control of mTORC1 in HCT116 cells was attributed to regulation of lysosomal dynamic of TSC2 and Rheb but not Sesn2 activity.

9) The data in Figure 7 are inconsistent. The number and size of colonies look similar between sh-Luc and sh-Sesn2 in Figure 7 and quite different in Figure 7. Moreover, it is not clear how the Figure 7 is relevant to the other parts of the paper. Does 5-Fu and CPT-11 treatments affect mTORC1 activity? Is mTORC1 responsible for the potential effect of Sesn2 on colony formation?

10) Figure 7: Despite the observed phenotypes on colon morphology, it is not clear which mechanisms are responsible for these phenotypes: whether they involve ER stress and mTORC1 regulation or both. To study whether inhibition of inflammation and colon carcinogenesis by Sesn2 is mediated by mTORC1 the authors should verify that the effects observed in Sesn2-null cells can be reversed by treatment with mTORC1 inhibitors.

11) If Sestrin level were modulated during colitis, the authors should examine the expression levels of Sestrins in DSS-induced colitis model. Are there any changes of Sestrins expression (both mRNA and protein level) before and after DSS treatment in mice?

---

## [Author Response]

*The reviewers discussed the manuscript and decided to encourage resubmission of a revised manuscript addressing several major concerns: 1) Figure 1: It is not clear where these samples were obtained from and how they were analyzed. What was used as a control? Although there was a trend for mRNA increase of Sesn2 and Sesn3 in patients (n=10), the error bar is pretty big, which makes the data weak. If the authors really want to make this point, more patients samples are needed.*

The samples in Figure 1 are colon samples from ulcerative colitis patients. The normal tissue is adjacent, non-inflamed tissue. These tissues were histologically confirmed and described previously (Gastroenterology 145:831-41). We have clarified this information and cited proper references in the revised manuscript.

In addition, we realized that the presented data are somewhat misleading because the error bars in Figure 1 of the original manuscript represented standard deviation (SD) in patient samples, which is large in nature. Specifically, SD does not reflect the power of the sample size; therefore, analyzing more patients would not help reduce this error bar. With n=10 per group, which is a reasonable sample size, we were able to detect strong statistical significance in the changes in expression of Sestrin2 and Sestrin3 between control and ulcerative colitis groups (*P*<0.01 and *P*<0.05, respectively), and error bars made from the standard error of the mean (SEM) did not overlap between these samples (Figure 1).

To eliminate the possibility of this potential confusion, we have used SEM error bars, which do reflect the power of the sample size, throughout the manuscript. We also wanted to note that we have eliminated the Crohn’s disease group from Figure 1, because they did not show any statistically significant differences from control group samples.

*2) Figure 1: According to the working hypothesis, Sesn2 protects from ER stress and inflammation. Nevertheless, there is no significant difference between 2 mouse strains in body weight during first 7 days of treatment indicating that Sesn2 is not directly involved in control of inflammation and the observed phenotype might be a residual effect of Sesn2 inactivation. The effects of Sesn2 have to be assessed immediately after DSS treatment, and the impact of the DSS treatment on tissue morphology, regulation of Sesn2 protein expression, mTORC1 activity, ER stress levels and the inflammation should be addressed.*

We agree with the reviewers that Sestrin2 may not be directly involved in the control of inflammation. Indeed, our bone marrow transplantation results (Figure 2) suggest that Sestrin2 in epithelia, rather than in inflammatory cells, is more important for colon homeostasis during DSS insults. Therefore, we performed the reviewer’s suggested experiment by examining the colons of WT and *Sesn2*- KO mice right after the 7 days of DSS treatment (DSS – 7D ONLY; Figure 1—figure supplement 2). Consistent with the observation that DSS-induced body weight losses were not significantly different between WT and *Sesn2*-KO mice at this time point (Figure 1), there were no detectable changes in colon length (Figure 1—figure supplement 2), which quantitatively represents tissue damage. Histological degeneration (Figure 1—figure supplement 2), inflammatory gene expression (*TNFα*; Figure 1—figure supplement 2) and ER stress target gene expression (*Xbp1s*; Figure 1—figure supplement 2) also did not show any significant difference between the two groups.

According to these new results and the reviewer’s suggestion, we have clarified that Sestrin2 is more important in the recovery process, rather than in the initial response to injury, by rewriting text in the Abstract and subtitles.

*3) The mice used in Figure 1 and Figure 2 are different ages and are not comparable. It is not clear why the authors used 6-month old mice in Figure 1 rather than the younger mice as in Figure 2. Please clarify.*

As the reviewer indicated, Figure 1 was done with 6-month old mice, while Figure 2 was done with 4-month old mice (1 month after bone marrow transplantation). Both ages are considered to be within the same stage of development/aging, and the mice are age-matched within each experimental set.

The results of Figure 1 and Figure 2 should be interpreted in a separate context because the mice in Figure 2 have undergone bone marrow transplantation, which is a severe intervention involving a lethal dose of irradiation. We believe that this procedure made a larger impact than the age difference, and the severe procedure is the reason why the colitis-inducing effect of DSS is different between these two experiments. Still, both of these results solidly support our conclusion that Sestrin2 in the non-hematopoietic compartment is more important for protecting the colon from DSS-induced damage.

In addition, one of our cancer cohorts (n=7) demonstrated that one week of 2.5% DSS treatment killed all two-month-old *Sesn2*-KO mice within the following two weeks, but not WT mice of the same age (Figure 5). Therefore, the DSS sensitivity of Sestrin2-deficient mice does not appear to be age-dependent.

*4) In Figure 2/F only DSS-treated mice are analyzed. To prove their claim the authors should compare the levels of ER stress in untreated and DSS-treated mice analyzing Sesn2 expression and the expression of the ER markers. Moreover, the protein extracts were not well normalized in Figure 2 and the actin marker is overexposed. Also the role of Sesn2 and ER stress in regulation of cell death is not addressed directly and not quantitated.*

As the reviewer suggested, we have analyzed the *Sesn2* expression and ER stress signaling in mice during the course of DSS treatment. Consistent with what was formerly reported (Gastroenterology 144:989-1000), DSS treatment induced prolonged ER stress signaling (monitored by expression of *Xbp1s* and other ER stress target genes; Figure 2—figure supplement 1) that was accompanied by modestly increased Sestrin2 expression in the recovery phase (Figure 2—figure supplement 1). Although the actin blot of Figure 2 was densely exposed, the protein loading was normalized according to the total protein concentration, and non-specific bands in the full image of the same blots (e.g. see CHOP blots in Figure 2—figure supplement 3) indicate that the general protein loading was normalized well. In the revised manuscript, we have included additional immunoblot images, in which the actin marker was more lightly exposed (Figure 2—figure supplement 2). In addition, analyses of qRT-PCR data, which was quantitatively normalized by *β-actin* mRNA (Figure 2), further confirmed that ER stress is indeed increased in *Sesn2*-KO mice when compared to WT mice. Quantification of colon length (Figure 1) unequivocally demonstrated that colons of *Sesn2*-KO mice do not recover from DSS treatment and remain severely damaged. However, it is beyond the scope of our research to determine whether ER stress directly regulates apoptotic cell death or indirectly induces tissue degeneration through disruption of colonic homeostasis.

*5) Figure 3. The subtle difference in mRNA levels between normal tissue and in tumors can be explained by compositions of different cell types in normal and tumor samples. Surprisingly, there is no significant downregulation of Sesn2 expression during cancer progression. Did the authors carry out an analysis of Sesn2 locus to find if Sesn2 locus has undergone LOH in tumors? Also related to this figure, unlike TP53, which is mutated in about half of all colorectal cancers, SESN2 is rarely mutated. The authors mined several public datasets demonstrating that SESN2 expression is lower in tumors relative to normal tissues. Is this lower expression in human tumors due to mutation or loss of p53? How is SESN2 expression between tumors expressing wild type versus mutant p53? Reanalysis of the various datasets in Figure 3 partitioning tumors based on p53 status would be a significant addition to the paper and overall model. Since the p53 field is heavily debating which of tis transcriptional targets carries tumor suppressive activity, this would be a good contribution to the field.*

We wanted to stress that the Sestrin2 mRNA difference between normal colon and tumors is not subtle, considering that these data are collected from a variety of different human samples. In many different analyses, the magnitude of Sestrin2 suppression in tumors is among the top 1-5% of all suppressed genes (Figure 5). The extent of the difference was also exceptionally strong (all studies indicate *P*<10_-4_ or much lower; in TCGA analysis, *P*=1.6 x 10_-30_). None of the other cell type markers, such as Villin (VIL1, enterocytes), DLL1 (progenitor cells), F4/80 (EMR1, macrophages), Gr-1 (LY6G5B, leukocytes), or β-catenin (CTNNB), showed such strong changes in expression as observed for Sestrin2 (Figure 3—figure supplement 1). Only some stem cell markers, such as Lgr5, were upregulated (not downregulated) in tumor samples (Figure 3—figure supplement 1). These new analyses were included in the revised manuscript.

Regarding the LOH, we did perform a copy number analysis for the *Sesn2* locus; it appeared that copy number of the Sestrin2 locus generally, and significantly, decreases with cancer progression (Figure 3). However, the extent of copy number loss was very small (~10%), suggesting that transcriptional downregulation (which is through p53 loss – see below), rather than the LOH, is the major mechanism of Sestrin2 loss during colon cancer progression.

It is a very insightful suggestion to analyze the correlation between p53 status and Sestrin2 expression in human colon cancer samples. We were able to perform this analysis using the cancer genome atlas (TCGA) dataset, as it has comprehensive information regarding the genomic status of each tumor, determined by whole genome/exome sequencing. From this database, we were able to classify all the colon tumor samples into the two groups. One group (designated as “p53-mutated”) has one or more missense or nonsense coding sequence mutations in the *p53* gene. The other group does not reveal any coding sequence mutations in the *p53* gene; however, it is still possible that these tumors contain *p53* mutations in essential non-coding regions (e.g. promoters, enhancers or introns) or other genomic or epigenetic alterations that can lead to functional p53 inactivation (e.g. MDM2 overexpression). Therefore, this second group was designated as “p53-unknown”.

Expressions of *SESN2* and other known p53 target genes were analyzed in three different groups: normal colons, “p53-unknown” tumors and “p53-mutated” tumors.

The analyses gave us very interesting results. First, we found that *SESN2* expression is significantly reduced in “p53-mutated” tumors compared to “p53-unknown” tumors (Figure 4), suggesting a role of p53 in controlling *SESN2* expression. This difference was consistently observed in three independent *SESN2* probes (Figure 4). Strikingly, there were almost no overlaps of *SESN2* expression levels between the normal colon and “p53-mutated” tumor groups (Figure 4), while *SESN2* levels in “p53-unknown” tumors overlap with both groups (Figure 4). None of the other known p53 target genes, such as *p21, GADD45A* and *MDM2*, showed this strong correlation: for these genes, considerable overlaps were found between the normal colon and “p53-mutated” tumor groups in individual samples (Figure 4). Nevertheless, expression levels of these genes have a general positive correlation with expression of *SESN2* in individual samples (Figure 4), suggesting that *SESN2, p21, GADD45A* and *MDM2* are all regulated through the same p53-dependent mechanism. Indeed, all of these genes are differentially expressed between “p53-unknown” and “p53-mutated” tumor groups (Figure 4 and Figure 4—figure supplement 1).

Interestingly, some p53 target genes, such as apoptosis mediators *BAX, PUMA* and *p53AIP1* or mTORC1 regulators *TSC2, AMPKβ* and *PTEN*, did not show any differential expression between the “p53-unknown” and “p53-mutated” tumor groups (Figure 4—figure supplement 1), suggesting that the effects of p53 mutations on these genes are minimal in the pathological context of colon carcinogenesis.

These results collectively highlight *SESN2* as a clinically relevant target of p53 during colon carcinogenesis. As the reviewer indicated, there has been a heavy debate in the p53 field about which of its transcription targets is important for its tumor suppressive activity. As we have demonstrated that *SESN2* is one of the most significant p53 targets in colon carcinogenesis, we agree with the reviewer that this is a very important contribution to the field.

*6) The experiments on cancer are not compatible with the experiments on colitis as different conditions were used for these experiments. What is the mouse age? Does Sesn2 protects from colitis, ER stress and cell death in the mice treated with 1.5% DSS in the colitis study. It is also not clear how tumor sizes and tumor load were analyzed.*

As indicated in our original manuscript, the colitis condition (3% or 2.5% DSS) cannot be used for the tumor experiment because it killed the *Sesn2*-KO mice. Also, in the tumor study, we do multiple rounds of DSS treatment and allow the mice to age for an extended period, as it takes time for a tumor to develop. The timeline for both experiments begins with an AOM injection when the mice are 2 months (Figure 5) or 4 months (Figure 6, after bone marrow transplantation) of age. After the DSS treatments and tumor incubation periods (as in Figure 5), the mice in these studies reach the end of the experiment at around 6 months (Figure 5) or 8 months (Figure 6), at which point the analyses were performed. Therefore, regardless of the DSS concentration, the DSS colitis experiment and the AOM/DSS experiment should be considered as separate experiments.

The mechanisms underlying increased susceptibility to colitis and colon cancer may be different from each other. For example, resolution of ER stress and subsequent recovery from DSS insult may be important for the increased colitis phenotype of *Sesn2*-KO mice, while Sestrin2-dependent suppression of mTORC1 and cell growth could be more critical for increased susceptibility of *Sesn2*-KO mice to colon cancer development. In the revised manuscript, we have discussed these possibilities in the Discussion section.

In addition, we have clarified the methods of how we analyzed tumor sizes and tumor load in the cancer studies. A dissecting microscope (4x magnification) was used to assess the tumor number and size. Tumor size was defined as the mean of the two largest diameters measured with digital calipers. Tumor volume was derived from tumor size. Consistent with their histological appearance, a spherical shape was assumed for colon polyps, thus tumor volume = 4/3πr_3_, where r = radius. Tumor burden/load is defined as the total polyp volume per animal, that is the product of polyp number and polyp volume.

*7) In Figure 5 there is no evidence that Sesn2 expression and mTORC1 activity is somehow affected in tumor samples in control mice. Also the data on 4E-BP1 staining look odd because several differently phosphorylated forms of 4E-BP1 usually appear when cells are stained with 4E-BP1 antibody. Moreover, to analyze mTORC1 activity S6K phosphorylation data should be added to the experiment and several samples should be presented.*

As discussed in our former manuscript, mouse AOM/DSS colon tumors do not typically lose p53 (Cancer Res 64: 6394-6401, Cancer Sci 95: 475-480), unlike what is seen in human colon cancers. Therefore, it is expected that Sestrin2 expression and mTORC1 activity are not strongly affected in tumor samples of control mice (Figure 6). However, we have demonstrated that p53 loss in mouse colon tumor can also inhibit Sestrin2 expression (Figure 7) as observed in human colon cancers (Figure 4).

The PAGE gel resolution of mouse tissue samples (especially the colon samples) is relatively poor (see the whole blots presented in supplemental figures). For this reason, the phosphorylation-induced 4E-BP1 shifts are not evident from this specific western blot (Figure 6). We do see clear 4E-BP1 shifts in cultured cell lysates (see Figure 7). In the revised manuscript, we have shown additional immunostaining (Figure 6—figure supplement 2) and immunoblot (Figure 6—figure supplement 3) results to demonstrate that these changes are reproducible in tissues from different mice. The 4E-BP1 shifts were visible in some of these additional blots, although with a lower resolution when compared to the results from cultured cell lysates.

We agree that S6K is another good readout of mTORC1 activity; however, for an unknown reason, the antibody worked very poorly in these mouse colon samples due to the presence of strong non-specific bands. Still, S6 and 4E-BP1 phosphorylation data (in both immunoblotting and immunostaining) consistently support that mTORC1 is strongly upregulated in Sestrin2-deficient mouse tumors. According to recent studies about mTORC1 signaling in colon (Nature (2015) 517:497-500; J Clin Invest (2013) 123:767-781; PLoS One (2014) 9:e96023), p- S6 and p-4E-BP1 antibodies are reliable reporters of mTORC1 signaling activity in colon tissue. These studies did not include p-S6K analyses.

*8) Figure 6. It was recently published by Agarwal et al., Mol. Can. Res 2015 that p53 inactivation in colon cancer cells led to strong suppression of Sesn2 expression as well as expression of some other p53-dependent mTORC1 regulators such as TSC2. Moreover, the control of mTORC1 in HCT116 cells was attributed to regulation of lysosomal dynamic of TSC2 and Rheb but not Sesn2 activity.*

We and others have recently shown that Sestrin2 controls lysosomal localization of mTORC1 through modulation of the GATOR complexes (Sci Rep. 2015 5:9502; Cell Rep. 2014 9:1; Cell Rep. 2014 9:1281). In addition to this mechanism, Sestrin2 controls the TSC2-Rheb axis through AMPK activation (Cell 2008 134:451; Science 327:1223). Therefore, the Agarwal et al. findings that p53 affects Rheb signaling do not contradict our conclusion that Sestrin2 controls mTORC1 signaling as a p53 target.

In response to the reviewer’s comment, we made a note in the revised manuscript that p53 has other possible targets that can control mTORC1 signaling, such as *TSC2, AMPKβ* and *PTEN* (Cancer Res 67:4043-53). However, these targets were not differentially expressed between “p53-unknown” and “p53-mutated” colon cancer tissues (Figure 4—figure supplement 1), while *SESN2, p21, GADD45* and *MDM2* are all differentially regulated between these groups (Figure 4 and Figure 4—figure supplement 1). We have included these new analyses in the revised manuscript.

*9) The data in Figure 7 are inconsistent. The number and size of colonies look similar between sh-Luc and sh-Sesn2 in Figure 7 and quite different in Figure 7. Moreover, it is not clear how the Figure 7 is relevant to the other parts of the paper. Does 5-Fu and CPT-11 treatments affect mTORC1 activity? Is mTORC1 responsible for the potential effect of Sesn2 on colony formation?*

In the revised manuscript, we have addressed this issue by repeating both experiments in a single batch, and replacing the old figures (Figure 7—figure supplement 2, in the revised manuscript). The new results are consistent with the former results and clearly show that sh-Sesn2 cells grow faster and are more resistant to 5-FU and CPT-11 treatments.

We have also examined Sestrin2 expression and mTORC1 signaling in RKO cells treated with 5-FU and CPT-11. After the 5-FU or CPT-11 treatments, Sestrin2 expression was slightly elevated in WT cells (sh-Luc; Figure 7—figure supplement 2). Interestingly, mTORC1 signaling, monitored by p-S6K and p-4E-BP1 was prominently upregulated in sh-Sesn2 cells after 5-FU treatment (Figure 7—figure supplement 2), suggesting that Sestrin2 functions to suppress mTORC1 activation after 5-FU treatment. In contrast, CPT-11 reduced mTORC1 signaling, and Sestrin2 silencing led to persistent mTORC1 upregulation (Figure 7—figure supplement 2). This mTORC1 upregulation as a result of Sestrin2 loss could have conferred chemoresistance to RKO cells against 5- FU and CPT-11.

Nevertheless, we agree with the reviewers that, although these results are interesting and potentially important in the field of tumor chemoresistance, they are peripheral to the main story. Therefore, we have removed these data from the main figure and moved them into the figure supplement (Figure 7—figure supplement 2).

*10) Figure 7. Despite the observed phenotypes on colon morphology, it is not clear which mechanisms are responsible for these phenotypes: whether they involve ER stress and mTORC1 regulation or both. To study whether inhibition of inflammation and colon carcinogenesis by Sesn2 is mediated by mTORC1 the authors should verify that the effects observed in Sesn2-null cells can be reversed by treatment with mTORC1 inhibitors.*

We have shown that increased colony growth phenotypes of either Sestrin2-silenced or p53-mutated (SW480; therefore Sestrin2-downregulated) cells can be suppressed by treatment with rapamycin, an mTORC1 inhibitor (Figure 7—figure supplement 1). Therefore, the role of Sestrin2 in inhibiting colon tumor growth seems to be mainly dependent on mTORC1 suppression.

11) If Sestrin level were modulated during colitis, the authors should examine the expression levels of Sestrins in DSS-induced colitis model. Are there any changes of Sestrins expression (both mRNA and protein level) before and after DSS treatment in mice?

We have shown that Sestrin2 level, in both mRNA and protein levels, was moderately increased after DSS treatment (Figure 2—figure supplement 1). This observation is consistent with the findings from the human colitis samples (Figure 1).